# SUPERVISED DIMENSION CONTRASTIVE LEARNING

## ABSTRACT

Representation learning is a fundamental task in machine learning, with the learned representations often serving as the backbone for downstream applications. While self-supervised learning has demonstrated strong generalization by maximizing representation diversity, it lacks explicit semantic structure. In contrast, supervised contrastive learning improves in-domain performance by clustering same-class representations but often limits diversity, reducing out-domain generalization. To address this, we redefine supervised representation learning from a mutual information perspective, highlighting the need to balance representation diversity and class relevance. We propose Supervised Dimension Contrastive Learning (SupDCL), a comprehensive framework that optimizes this balance through three key components: (1) decorrelation loss to enhance representation diversity, (2) orthogonal loss to remove redundant information, and (3) class correlation loss to strengthen class alignment. SupDCL achieves state-of-the-art generalization across ImageNet-1K and 10 downstream tasks, bridging the gap between self-supervised and supervised learning. By optimizing mutual information, it provides a principled approach to supervised representation learning, ensuring representations that are both robust and transferable.

## 1 INTRODUCTION

Recent advances in Self-Supervised Learning (SSL) (Chen et al., 2020; Zbontar et al., 2021; Caron et al., 2020; 2021) have demonstrated its effectiveness as a pretraining strategy using large-scale unlabeled data. These methods leverage data augmentation to generate semantically similar pairs and align their representations. While SSL excels in learning diverse representations that generalize well to unseen distributions, as shown in the red region of Figure 1, its reliance on data augmentation and instance discrimination often limits the ability to precisely differentiate positive samples from hard negatives (Robinson et al., 2020; Wu et al., 2020). This trade-off results in high out-domain performance but suboptimal in-domain accuracy, particularly for tasks requiring semantic structure.

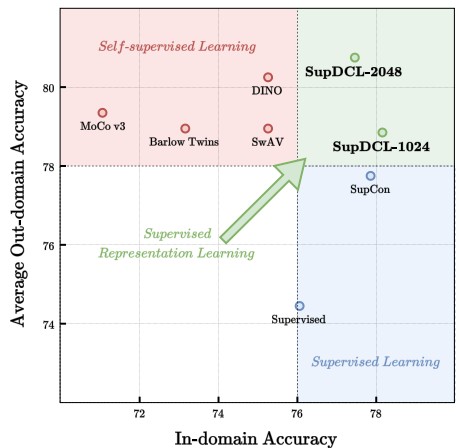

Figure 1: In-domain and out-domain accuracy (%) of models pretrained on ImageNet-1K using various learning methods.

With the growing availability of large labeled datasets such as JFT-300M (Sun et al., 2017), Supervised Representation Learning (SRL) has gained attention for its ability to leverage labels to capture semantically meaningful relationships. Supervised contrastive learning methods (Khosla et al., 2020; Zha et al., 2024; Cui et al., 2021; 2023) extend SSL's principles by clustering representations of the same class while separating those of different classes. These methods have demonstrated strong in-domain performance by effectively minimizing intra-class variability, as evident in the blue region of Figure 1. However, this tight clustering comes at the expense of representation diversity, leading to weaker generalization to out-domain tasks. Such limitations arise because these methods primarily focus on class-wise clustering without addressing the need to retain diverse and fine-grained information within each class.

To address these challenges, we revisit SRL through a mutual information perspective and redefine it as the process of maximizing the mutual information between representations $Y$ and their supervision signals $C$. Specifically, $I(Y; C)$ can be decomposed into two key terms:

1. $H(Y)$: Total information content of the representation, which ensures diversity and richness.

2. $H(Y|C)$: Redundant or irrelevant information that is not related to the supervision signal.

An effective SRL framework must maximize $H(Y)$ to retain diverse information while minimizing $H(Y|C)$ to ensure class relevance. However, existing approaches fail to balance these two objectives, resulting in either high in-domain performance with poor generalization (supervised methods) or strong generalization with reduced semantic alignment (self-supervised methods).

To overcome these limitations, we propose Supervised Dimension Contrastive Learning (SupDCL), a novel and comprehensive framework designed to balance representation diversity and class discriminability. SupDCL introduces three core objectives:

1. Decorrelation loss to enforce statistical independence between feature dimensions, thereby maximizing $H(Y)$.

2. Orthogonal loss to filter out redundant information and ensure task-relevant features are effectively captured.

3. Class correlation loss to align representations with semantic supervision, minimizing $H(Y|C)$.

As shown in the green region of Figure 1, SupDCL achieves a balance between in-domain and out-domain performance, bridging the gap between self-supervised and supervised methods. While self-supervised methods like SimCLR (Chen et al., 2020) and Barlow Twins (Zbontar et al., 2021) excel in generalization by maximizing $H(Y)$, and supervised methods like SupCon (Khosla et al., 2020) achieve high in-domain accuracy by minimizing $H(Y|C)$, SupDCL optimizes both terms simultaneously. This balance allows SupDCL to achieve state-of-the-art generalization performance across diverse benchmarks.

Through extensive experiments on ImageNet-1K and 10 downstream tasks, we demonstrate that SupDCL not only outperforms existing supervised and self-supervised approaches but also provides a principled framework for representation learning based on mutual information.

## 2 RELATED WORKS

### 2.1 SELF-SUPERVISED LEARNING

Self-supervised learning (SSL) has emerged as a powerful paradigm for learning meaningful representations from unlabeled data, demonstrating strong performance across various downstream tasks (Chen et al., 2020; He et al., 2020; Caron et al., 2020; Grill et al., 2020; Chen & He, 2021; Caron et al., 2021; Zbontar et al., 2021; Bardes et al., 2021). Unlike supervised learning, SSL does not rely on human-labeled annotations, making it scalable but inherently agnostic to semantic class structures. Various approaches have been proposed to address this challenge, primarily categorized into contrastive, non-contrastive, and dimension contrastive learning.

Contrastive learning methods such as SimCLR (Chen et al., 2020) and MoCo (He et al., 2020) define positive pairs from augmented views of the same image and negative pairs from different images. These methods enforce instance discrimination, maximizing representation diversity but requiring large batch sizes or memory banks to maintain sufficient negative samples.

Non-contrastive learning eliminates the need for negatives and instead relies on architectural constraints to prevent representation collapse. BYOL (Grill et al., 2020) and SimSiam (Chen & He, 2021) use asymmetric predictor networks, while DINO (Caron et al., 2021) applies knowledge distillation to align representations across different network views. These methods implicitly optimize representation diversity

Dimension contrastive learning shifts the focus from instance-level relationships to decorrelating embedding dimensions to increase information content (Zbontar et al., 2021; Bardes et al., 2021). Barlow Twins (Zbontar et al., 2021) minimizes cross-correlation between dimensions to enforce statistical independence, while VICReg (Bardes et al., 2021) jointly optimizes invariance, variance, and covariance objectives. These methods suggest that independent embedding dimensions facilitate richer representation learning, aligning with our study on entropy-based representation analysis.

While SSL excels in capturing diverse representations, its lack of direct semantic supervision can lead to representations that are not well-aligned with human-defined categories. Our work builds upon these insights by incorporating supervised information while maintaining the advantages of dimension-level independence, ensuring both diversity and class relevance.

## 2.2 SUPERVISED LEARNING

Supervised learning typically relies on cross-entropy loss, which optimizes classification performance but does not explicitly encourage representation diversity (Elsayed et al., 2018; Cao et al., 2019; Zhang & Sabuncu, 2018). While techniques such as label smoothing (Müller et al., 2019; Szegedy et al., 2016) and augmentation-based methods like MixUp (Zhang et al., 2017) aim to improve robustness, they do not directly optimize the structure of learned representations.

Supervised learning methods commonly rely on cross-entropy loss for classification tasks. However, its limitations, including sensitivity to class imbalance and reliance on instance-specific predictions, are well-documented (Elsayed et al., 2018; Cao et al., 2019; Zhang & Sabuncu, 2018). Studies propose alternatives involving label distribution modifications (Müller et al., 2019; Szegedy et al., 2016) and advanced augmentations like MixUp (Zhang et al., 2017).

Supervised contrastive learning (SupCon) (Khosla et al., 2020) extends contrastive methods by defining positive pairs using same-class samples, effectively minimizing intra-class variance and maximizing inter-class separation. This approach strengthens semantic consistency but inherently limits representation diversity since the number of unique representations is constrained by the number of classes.

Recent variants such as PaCo (Cui et al., 2021) and GPaCo (Cui et al., 2023) introduce class-wise learnable centers, further improving robustness under class imbalance. However, these methods still do not explicitly control redundancy across representation dimensions, which can limit overall information content.

Our method, Supervised Dimension Contrastive Learning (SupDCL), extends these ideas by explicitly balancing representation diversity $H(Y)$ and class relevance $H(Y|C)$.

## 3 SUPERVISED REPRESENTATION LEARNING

Unlike conventional supervised learning, which focuses on task-specific performance, Supervised Representation Learning (SRL) aims to learn representations that capture rich and structured information beyond a single task. Given an input $X$, its representation $Y$ is obtained via an encoder $f$ and is optimized to retain relevant information from the supervision signal $C$. This can be formulated as maximizing the mutual information between $Y$ and $C$:

$$I(Y; C) = H(Y) - H(Y|C). \quad (1)$$

For effective SRL, two key objectives must be balanced:

1. Maximizing $H(Y)$ to ensure the representation is diverse and information-rich.

2. Minimizing $H(Y|C)$ to remove information that is not semantically relevant to supervision.

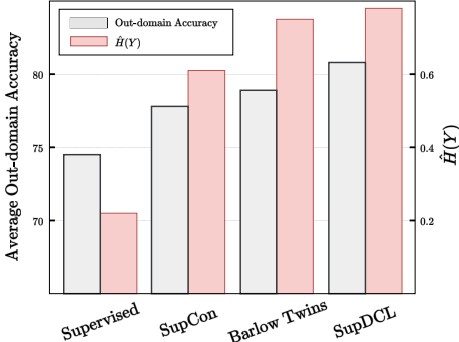

Figure 2: Comparison of relative representation entropy approximation $\hat{H}(Y)$ and average out-domain Top-1 accuracy (%) across different methods. Models are pretrained on ImageNet-1K, and out-domain performance reflects their generalization capability.

By jointly optimizing these objectives, SRL enhances not only in-domain performance but also generalization and transferability across different domains and tasks.

A previously studied SRL approach, Supervised contrastive learning (SupCon), enforces within-class clustering, ensuring that representations of the same class remain close in the embedding space. This formulation explicitly conditions $Y$ on $C$, reducing intra-class variability and thereby minimizing

$H(Y|C)$. Class-centric clustering inherently restricts the number of distinct representations to the number of classes, which in turn constrains $H(Y)$ and limits $I(Y; C)$. While this framework enhances in-domain accuracy, it limits representation diversity and hinders the ability to generalize to unseen distributions, as shown in Figure 1.

Self-Supervised Learning (SSL) methods such as SimCLR and Barlow Twins maximize $I(X; Y)$ by enforcing instance discrimination. SSL ensures that each input maps to a distinct representation and $H(Y)$ increases with the dataset size. However, the absence of supervision prevents minimizing $H(Y|C)$. The resulting representations maintain diversity but lack semantic structure, which is reflected in the limited in-domain performance shown in Figure 1.

To empirically validate this approach, we analyze the relationship between representation diversity $H(Y)$ and out-domain generalization across various methods. Measuring the Shannon entropy of high-dimensional representations directly is impractical due to data and computational requirements. To address this, we analyze representation diversity post hoc using a surrogate metric based on the spectral structure of feature activations.

Specifically, we follow prior work (Ansuini et al., 2019) that estimates the PCA based intrinsic dimensionality (PC-ID), defined as the number of principal components required to preserve a certain threshold of the total variance. This quantity approximates how broadly the variance is distributed across dimensions. We use it as a proxy for the degree of dispersion in the learned representations.

In our work, we set the threshold to 95% and normalize by the full feature dimension D, defining the metric as:

$$\hat{H}(Y) \triangleq \frac{d}{D}, \quad \text{s.t.} \quad \frac{\sum_{i=1}^{d} \lambda_i}{\sum_{i=1}^{D} \lambda_i} \geq 0.95, \tag{2}$$

where $\lambda_i$ are the eigenvalues of the representation covariance matrix, $D$ is the full feature dimension, and $d$ is the number of components needed to capture 95% of the total variance.

While $\hat{H}(Y)$ is not a rigorous information-theoretic entropy, it captures how broadly variance is dispersed across feature dimensions. Under a Gaussian assumption, entropy can be expressed in terms of the covariance spectrum, which motivates interpreting PC-ID as a surrogate diversity measure.

However, learned representations are generally non-Gaussian and may reside on nonlinear manifolds. To complement PC-ID, we therefore evaluate additional diversity estimators, including the TwoNN intrinsic dimensionality, which does not rely on linear or Gaussian assumptions, as well as spectrum-based measures such as the log-determinant. These complementary metrics provide alternative views of entropy and representation dispersion, and they exhibit consistent trends with PC-ID. Full results are presented in Appendix A

As shown in Figure 2, we observe that models with higher $\hat{H}(Y)$, such as SupDCL and Barlow Twins, tend to generalize better across domains. This suggests that promoting diverse yet structured representations contributes to robustness under distribution shifts, emphasizing the importance of jointly maximizing $H(Y)$ while preserving class relevance via low $H(Y|C)$.

## 4 SUPERVISED DIMENSION CONTRASTIVE LEARNING

We propose a novel supervised representation learning method based on Dimension Contrastive Learning (DCL) (Zbontar et al., 2021; Bardes et al., 2021; Garrido et al., 2022). DCL directly treats each dimension of the learned representation as an information carrier. Our approach explicitly aligns each dimension of the representation $Y$ with the supervision $C$ (e.g., class label) to enhance the quality of the learned representation.

Our method integrates supervision to ensure individual dimensions in $Y$ encode supervision-relevant information. We introduce a learnable aggregator $h : Y \rightarrow \hat{C}$ that extracts class-relevant information from $Y$, generates a class prediction $\hat{C}$, and serves as an explicit information selector. $h$ makes representation $Y$ effectively encode supervision-relevant information while filtering out irrelevant features.

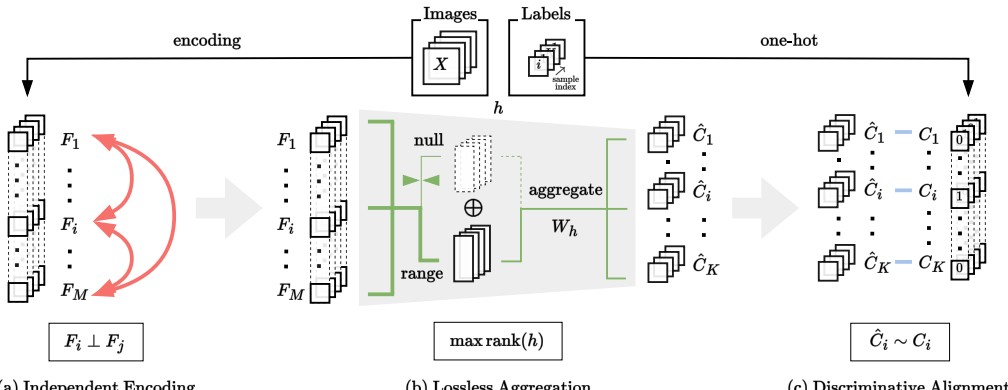

(a) Independent Encoding    (b) Lossless Aggregation    (c) Discriminative Alignment

Figure 3: **Supervised Dimension Contrastive Learning (SupDCL) (a) Independent Encoding**: Decorrelation loss increases the representation entropy $H(Y)$ by making each dimension $F_i$ independent, reducing redundancy among dimensions. **(b) Lossless Aggregation**: Orthogonal loss maximizes the rank of the aggregator $h$, preventing information loss during aggregation and reducing irrelevant information $H(Y_{\text{null}}|\hat{C}, C)$. **(c) Discriminative Alignment**: Class Correlation loss aligns the predicted class variables $\hat{C}$ with the true class variables $C$, minimizing uncertainty $H(\hat{C}|C)$. By combining these processes, SupDCL encourages higher mutual information $I(Y; C)$ for supervised representation learning.

The representation $Y$ is decomposed using the pseudo-inverse of the weight matrix $W_h$ of the aggregator $h$, which is assumed to be a linear operator for ease of derivation:

$$Y = Y_{\text{range}} \oplus Y_{\text{null}}, \tag{3}$$

$$\text{where} \quad Y_{\text{range}} = W_h^{\dagger} W_h Y, \quad Y_{\text{null}} = (I - W_h^{\dagger} W_h) Y. \tag{4}$$

$Y_{\text{range}}$ represents the information used by the aggregator $h$ with weight $W_h$ when making the class prediction $\hat{C}$, as it lies within the range space of $W_h$. $Y_{\text{null}}$ belongs to the null space of $W_h$, meaning it is mapped to zero when passed through $W_h$, as $W_h Y_{\text{null}} = W_h(I - W_h^{\dagger} W_h)Y = 0$, and does not affect the prediction. These components can decompose the supervised representation learning objective as follows.

**Proposition 4.1.** *Given a representation $Y$, class information $C$, and a linear aggregator $h(Y) = W_h Y = \hat{C}$, the mutual information between $Y$ and $C$ is decomposed as:*

$$I(Y; C) = H(Y) - H(Y_{null}|\hat{C}, C) - H(\hat{C}|C), \tag{5}$$

*where $Y_{null} = (I - W_h^{\dagger} W_h)Y$.*

*Proof.* Proof is provided in the Appendix B. ☐

Each term represents a distinct factor influencing the supervised representation learning process:

1. $H(Y)$: The total information content of the representation, which should be maximized to ensure a rich and informative representation.

2. $H(Y_{\text{null}}|\hat{C}, C)$: The amount of information in $Y$ that is neither captured by the aggregator $h$ nor useful for class prediction. This term should be minimized to remove irrelevant or redundant information.

3. $H(\hat{C}|C)$: The uncertainty in class prediction $\hat{C}$ given the true class $C$. This term should be minimized to ensure the representation is discriminative for the given task.

This decomposition provides a principled framework for optimizing each of these three aspects independently, enabling effective learning of supervised representations. Building on this perspective, we design our loss functions under the linear assumption within the dimension contrastive learning framework (Zbontar et al., 2021; Bardes et al., 2021; Garrido et al., 2022), such that each proposed objective corresponds to one component of the decomposition of $I(Y; C)$.

## 4.1 INDEPENDENT ENCODING FOR $\max \mathbf{H}(\mathbf{Y})$

To maximize the representation entropy $H(Y)$, we introduce decorrelation loss $\mathcal{L}_{\text{decorr}}$. This makes each dimension $F_i$ in $Y$ independently capture non-redundant information. This approach follows Barlow Twins and VICReg, where Pearson correlation reduces inter-dimensional dependency within the representation. The Pearson correlation between two dimensions $F_i$ and $F_j$ of $Y$ is given by:

$$\text{Corr}(F_i, F_j) = \frac{\text{Cov}(F_i, F_j)}{\sqrt{\text{Var}(F_i)\text{Var}(F_j)}}, \tag{6}$$

where Cov and Var denote the empirical covariance and variance computed from batch data, respectively. Then, the decorrelation loss is defined as:

$$\mathcal{L}_{\text{decorr}} \triangleq \sum_i \sum_{j \neq i} [\text{Corr}(F_i, F_j)]^2. \tag{7}$$

$\mathcal{L}_{\text{decorr}}$ enforces statistical independence among representation dimensions, reducing redundancy and thereby maximizing the total information content in $Y$.

## 4.2 LOSSLESS AGGREGATION FOR $\min \mathbf{H}(\mathbf{Y}_{\text{NULL}}|\hat{\mathbf{C}}, \mathbf{C})$

To minimize the uncollected redundant information $H(Y_{\text{null}}|\hat{C}, C)$ in the representation $Y$, we introduce an orthogonal loss $\mathcal{L}_{\text{ortho}}$ that maximizes the capacity of the aggregator $h$ while eliminating unnecessary information. The class-related information in $Y_{\text{null}}$ is learned outside the influence of $h$, and information unrelated to $\hat{C}$ also lies beyond the direct control of $h$. Thus, it is necessary to reduce the information content of $Y_{\text{null}}$ itself.

Full rank preservation in $h$ constrains $Y_{\text{null}}$ dimensionality. The bounded dimensions naturally reduce the information content in $Y_{\text{null}}$. The orthogonal loss enforces linear independence in the weight matrices $W_h$ of $h$ through the following formulation:

$$\mathcal{L}_{\text{ortho}} \triangleq \sum_l \|(W_h^{(l)})^T W_h^{(l)} - I\|_F^2, \tag{8}$$

where $W_h^{(l)}$ denotes the weight matrix at $l$-th layer of $h$. The layer-wise orthogonality constraint maintains linear independence across all layers, effectively restricting the dimensionality of $Y_{\text{null}}$.

## 4.3 DISCRIMINATIVE ALIGNMENT FOR $\min \mathbf{H}(\hat{\mathbf{C}}|\mathbf{C})$

A class correlation loss $\mathcal{L}_{\text{corr}}$ aligns the distributions of predicted class $\hat{C}$ and ground truth $C$, minimizing uncertainty $H(\hat{C}|C)$. Following principles similar to decorrelation loss, the alignment utilizes empirical distributions within each batch and employs the same correlation metric to drive corresponding variable pairs toward unity:

$$\mathcal{L}_{\text{corr}} \triangleq \sum_i [1 - \text{Corr}(\hat{C}_i, C_i)]^2, \tag{9}$$

where $\hat{C}_i$ and $C_i$ represent the predicted and ground truth variables for the $i$-th class, respectively.

**Overall Loss Function** The overall loss function for SupDCL is defined as the combination of the three proposed loss terms:

$$\mathcal{L}_{\text{SupDCL}} \triangleq \lambda_{\text{decorr}}\mathcal{L}_{\text{decorr}} + \lambda_{\text{ortho}}\mathcal{L}_{\text{ortho}} + \lambda_{\text{corr}}\mathcal{L}_{\text{corr}}, \tag{10}$$

where $\lambda_{\text{decorr}}$, $\lambda_{\text{ortho}}$, and $\lambda_{\text{corr}}$ are scaling factors. These components are integrated to maximize the mutual information $I(Y; C)$ in a tractable manner (see Figure 3).

**Implementation Details** SupDCL follows the structure of SSL by introducing a projector $g : Y \rightarrow Z$, where the loss functions are computed in the embedding space $Z$. Additionally, augmentation invariance is incorporated to maintain consistency across different augmented views of the same data. For the invariance objective, SupDCL adopts the correlation-based loss function $\mathcal{L}_{\text{inv}}$ from Barlow Twins, promoting stable feature representations across augmentations. SupDCL adds minimal overhead compared to Barlow Twins, with only 3–9% more FLOPs depending on the backbone (see Appendix C). It is trained for 1000 epochs, which aligns with standard practice in prior self-supervised learning work.

Table 1: Top-1 accuracy (%) comparison for ResNet-50 pretrained on ImageNet-1K, evaluated on in-domain (ImageNet-1K) and out-domain (10 datasets). In-domain accuracy is reported for ImageNet-1K using a linear evaluation protocol. Out-domain accuracy is averaged across 10 datasets, and individual results are also listed.

| Method | In-domain IN1K | Out-domain | | | | | | | | | | |
|---|---|---|---|---|---|---|---|---|---|---|---|---|
| | | CIFAR10 | CIFAR100 | Food | Pets | Flowers | Caltech101 | Cars | Aircraft | DTD | SUN397 | Average |
| *Self-supervised Representation Learning:* | | | | | | | | | | | | |
| SimCLR | 69.1 | 90.6 | 71.6 | 68.4 | 83.6 | 91.2 | 90.3 | 50.3 | 50.3 | 74.5 | 58.8 | 73.0 |
| Barlow Twins | 73.2 | 92.9 | 78.3 | 76.1 | 89.9 | 97.7 | 89.9 | 65.4 | 60.2 | 76.9 | 62.9 | 79.0 |
| SwAV | 75.3 | 94.1 | 79.7 | 76.9 | 87.7 | 97.2 | 90.9 | 61.8 | 58.0 | 77.8 | 65.8 | 79.0 |
| MoCo v3 | 71.1 | **94.8** | **80.1** | 73.9 | 90.7 | 96.9 | **91.7** | 65.9 | 61.4 | 75.7 | 63.0 | 79.4 |
| VICReg | 73.2 | 92.7 | 78.2 | 76.1 | 89.4 | 97.9 | 89.6 | 66.7 | 60.5 | 77.7 | 62.6 | 79.1 |
| DINO | 75.3 | 93.9 | 79.4 | **78.6** | 89.3 | 97.8 | 90.9 | 67.9 | **62.4** | **77.2** | **65.9** | 80.3 |
| *Supervised Learning:* | | | | | | | | | | | | |
| Supervised | 76.1 | 91.7 | 74.4 | 71.2 | 92.3 | 95.4 | 88.7 | 50.0 | 48.6 | 71.9 | 60.4 | 74.5 |
| SupCon | 77.9 | 93.0 | 76.3 | 71.9 | 92.8 | 96.5 | **91.7** | 61.2 | 57.3 | 74.7 | 62.9 | 77.8 |
| PaCO | 78.7 | 91.1 | 70.6 | 64.4 | 92.3 | 88.4 | 87.9 | 37.8 | 34.8 | 68.1 | 58.2 | 69.4 |
| GPaCo | 79.5 | 92.2 | 73.5 | 62.5 | 91.9 | 84.4 | 88.4 | 37.6 | 32.7 | 67.7 | 57.2 | 68.8 |
| *Supervised Representation Learning:* | | | | | | | | | | | | |
| **SupDCL-1024 (Ours)** | 78.2 | 93.8 | 78.5 | 74.3 | **93.1** | 96.6 | **91.7** | 66.5 | 56.2 | 74.6 | 63.9 | 78.9 |
| **SupDCL-2048 (Ours)** | 77.5 | 94.1 | 79.9 | 78.3 | 92.6 | **98.1** | 91.3 | **71.1** | 61.9 | 75.9 | 64.7 | **80.8** |

## 5 EXPERIMENTS

**Baselines** We compare our method with self-supervised methods, including SimCLR (Chen et al., 2020), Barlow Twins (Zbontar et al., 2021), SwAV (Caron et al., 2020), MoCov3 (Chen et al., 2021), VICReg (Bardes et al., 2021), and DINO (Caron et al., 2021), as well as supervised methods using cross-entropy or contrastive learning such as SupCon (Khosla et al., 2020), PaCo (Cui et al., 2021), and GPaCo (Cui et al., 2023). We evaluate on in-domain classification and out-domain transfer tasks. While recent methods such as MAE (He et al., 2022), I-JEPA (Assran et al., 2023), and DINOv2/v3 (Oquab et al., 2023; Siméoni et al., 2025) leverage large-scale ViTs and masked image modeling, our evaluation is restricted to ResNet-50 backbones due to computational limits. We therefore include DINO (Caron et al., 2021) as a representative baseline, Since masked image modeling is complementary to our framework, its integration is left for future work.

**Datasets** For main evaluations, we pretrain a ResNet-50 backbone on ImageNet-1K and assess performance using a linear layer. Classification is evaluated using the standard linear evaluation protocol on ImageNet-1K, while transfer learning is assessed on 10 downstream tasks (CIFAR10/100 (Krizhevsky et al., 2009), Food (Bossard et al., 2014), Pets (Parkhi et al., 2012), Flowers (Nilsback & Zisserman, 2008), Caltech101 (Fei-Fei et al., 2004), Cars (Krause et al., 2013), Aircraft (Maji et al., 2013), DTD (Cimpoi et al., 2014), SUN397 (Xiao et al., 2010)) with the standard linear transfer protocol (Sun et al., 2017).

**Setup** In SupDCL, we use a 3-layer non-linear MLP for both the projector and aggregator. The embedding dimension is set to 1024 or 2048, corresponding to SupDCL-1024 and SupDCL-2048, which are trained on ImageNet-1K. Following Barlow Twins, we add an augmentation invariance loss (weight = 1). The decorrelation loss uses $\lambda_{\text{decorr}} = 0.04$ for 1024-dim embeddings and $\lambda_{\text{decorr}} = 0.02$ for 2048-dim embeddings with higher weight for lower dimensions to compensate for capacity limits. The other loss weights are fixed to $\lambda_{\text{ortho}} = 0.1$ and $\lambda_{\text{corr}} = 1$. For large-scale evaluation, we pretrain a ResNet-50 backbone on the ImageNet dataset using a batch size of 2048 with 4 H100 GPUs. We utilize the LARS optimizer with a weight decay of $1.5 \times 10^{-6}$, training for 1000 epochs with a learning rate of 0.2, which is linearly warmed over the first 10 epochs, followed by cosine decay. For all analyses and ablation studies, we pretrain a ResNet-18 backbone on CIFAR-100 using a single A6000 GPU. The model is trained with SGD, using a learning rate of 0.03, cosine scheduling with a 10-epoch warm-up, a weight decay of $5 \times 10^{-4}$, and a momentum of 0.9. The batch size is 256, and the embedding dimension is 128. Details on downstream tasks are provided in Appendix D.

### 5.1 MAIN RESULTS

SupDCL effectively balances in-domain and out-domain performance. As shown in Table 1, SupDCL-2048 attains the best out-domain accuracy, surpassing DINO, while SupDCL-1024 yields higher in-domain accuracy than SupCon with only a slight drop in out-domain generalization. We further analyze this trade-off in Section 5.3. By contrast, PaCo and GPaCo show strong in-domain results but degrade substantially under distribution shift, highlighting SupDCL's robustness across domains.

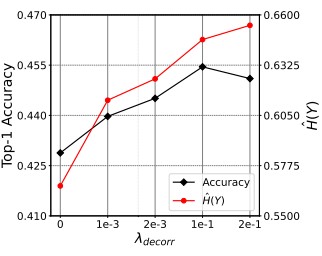 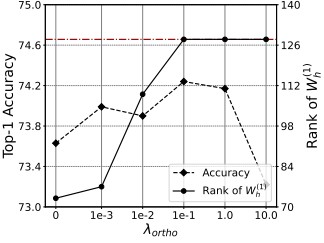 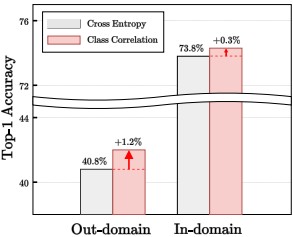

(a) Representation Entropy     (b) Aggregator Rank     (c) Alignment Effectiveness

Figure 4: (a) Representation Entropy with $\mathcal{L}_{\text{decorr}}$: Relative representation entropy approximation $\hat{H}(Y)$ and Top-1 accuracy (%) of SupDCL as a function of $\lambda_{\text{decorr}}$. (b) Aggregator Rank with $\mathcal{L}_{\text{ortho}}$: Rank of $W_h^{(1)}$ and Top-1 in-domain accuracy (%) of SupDCL as a function of $\lambda_{\text{ortho}}$. The maximum rank of $W_h^{(1)}$ is 128. (c) Alignment Effectiveness of $\mathcal{L}_{\text{corr}}$: Top-1 accuracy (%) comparison between cross-entropy and class correlation loss for in-domain and out-domain tasks.

## 5.2 ANALYSIS

**Decorrelation Loss** The effect of decorrelation loss on representation entropy is evaluated using the relative entropy approximation $\hat{H}(Y)$ and out-domain performance for different $\lambda_{\text{decorr}}$ values. Figure 4(a) shows that $\hat{H}(Y)$ increases consistently with higher $\lambda_{\text{decorr}}$. The results indicate that decorrelation loss functions as intended by enhancing the information content of the representation. A positive correlation between $\hat{H}(Y)$ and out-domain performance demonstrates that higher entropy contributes to improved generalization in out-domain tasks. However, large $\lambda_{\text{decorr}}$ values can lead to performance stagnation or reduced discriminative capacity. Uncontrolled entropy growth may introduce noise unrelated to the given supervision, requiring a balance between information diversity and effective learning of supervised signals.

**Orthogonal Loss** Figure 4(b) demonstrates the relationship between orthogonal regularization and the rank of the first layer of the aggregator $h$. Increasing the orthogonal loss weight $\lambda_{\text{ortho}}$ from 0 to 0.001 results in a higher rank of the aggregator, confirming the intended rank improvement through orthogonal regularization. As the rank increases, the entropy $H(Y_{\text{null}})$ decreases, suggesting a reduction of irrelevant information within $Y_{\text{null}}$. Consequently, this leads to improved in-domain performance by generating representations with reduced noise and enhanced task-relevance. However, increasing $\lambda_{\text{ortho}}$ beyond the point of achieving full-rank potentially degrades performance. The results demonstrate that the full-rankness of the aggregator is critical for in-domain performance.

**Class Correlation Loss** Class correlation loss is systematically compared to cross-entropy loss to assess its predictive capabilities. While both aim to predict class labels, their learning mechanisms differ significantly. Cross-entropy aligns predicted values $\hat{C}$ with true labels $C$ on a pointwise basis. This approach gives greater weight to regions with high density in the training data distribution $p_{\text{train}}(x)$, making it more sensitive to discrepancies between training and testing distributions. In contrast, class correlation loss aligns the empirical distributions of predicted class variables $\hat{C}$ and true class variables $C$ across a batch. By leveraging Pearson correlation for global alignment and applying normalization, it reduces the influence of data distribution shifts.

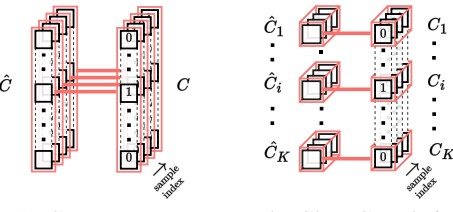

(a) Cross Entropy     (b) Class Correlation

Figure 5: (a) Cross Entropy: Aligns individual predictions $\hat{C}$ with true class labels $C$ on a per-sample basis. (b) Class Correlation: Aligns the predicted class variables $\hat{C}_i$ with true class variables $C_i$ by matching their empirical distributions across a batch.

Figure 5 illustrates the differences between the two loss functions, highlighting their learning approaches. As shown in Figure 4(c), class correlation loss outperforms cross-entropy in both in- and out-domain scenarios, with notable improvements in out-domain performance. The results demonstrate the robustness of correlation-based alignment to variations in data distribution.

Table 2: Top-1 accuracy (%) of in-domain performance under loss function ablation experiments.

| Method | $\mathcal{L}_{inv}$ | $\mathcal{L}_{decorr}$ | $\mathcal{L}_{ortho}$ | $\mathcal{L}_{cls}$ | Accuracy |
|---|---|---|---|---|---|
| No Supervision | ✓ | ✓ | - | - | 41.3 |
| SupDCL w/o $\mathcal{L}_{inv}$ | - | ✓ | ✓ | ✓ | 72.9 |
| SupDCL w/o $\mathcal{L}_{decorr}$ | ✓ | - | ✓ | ✓ | 73.3 |
| SupDCL w/o $\mathcal{L}_{ortho}$ | ✓ | ✓ | - | ✓ | 73.6 |
| SupDCL | ✓ | ✓ | ✓ | ✓ | **74.2** |

Table 3: Top-1 accuracy (%) of in-domain performance using various aggregator configurations.

| # Layers | Linearity | Accuracy |
|---|---|---|
| 2 | linear | 73.1 |
| 3 | linear | 74.1 |
| 3 | non-linear | **74.2** |
| 4 | non-linear | 74.1 |

## 5.3 ABLATION STUDY

**Loss Functions**  Table 2 presents the ablation study results for various loss functions in SupDCL. The study includes the proposed decorrelation loss, orthogonal loss, and class correlation loss, as well as an experiment where the augmentation invariance loss $\mathcal{L}_{inv}$ is excluded. The results show that the most critical factor for performance is the presence of supervision, as removing it leads to a significant performance drop to $41.3\%$. Each loss component contributes effectively, and performance decreases when any component is excluded. These findings highlight the importance of each component in SupDCL's design for enhancing performance.

**Aggregator**  We examine the effect of aggregator design by varying depth and non-linearity. As shown in Table 3, the 3-layer non-linear aggregator yields the best performance. Adding more layers does not further improve results, indicating that a moderately deep, non-linear design is sufficient for effec-

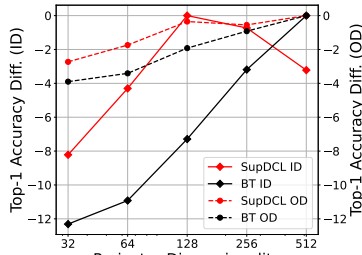

Figure 6: Top-1 accuracy difference for in-domain (ID) and out-domain (OD) performance analyzed across projector dimensionalities. Each line represents the difference from its respective maximum value, comparing SupDCL and Barlow Twins (BT).

tive feature integration. Moreover, gains from orthogonal regularization indicate its effectiveness beyond the linear aggregator assumption, including in non-linear MLP aggregators.

**Projector**  Figure 6 and Table 4 present the impact of the projector on performance. Figure 6 shows that increasing projector dimensionality improves out-domain performance due to decorrelation but introduces a trade-off for in-domain performance in SupDCL, as it relies on supervised signals. In contrast, Barlow Twins (BT) benefits from larger dimensions in both in- and out-domain performance due to its focus on capturing diverse information. Table 4 compares class correlation applied directly to the representation versus after passing through the projector. Applying it post-projector yields better performance, highlighting the projector's role in enhancing class correlation learning.

Table 4: Top-1 accuracy (%) of in-domain performance comparing the application of aggregation and class correlation on the representation or embedding space.

| Applied Space | Accuracy |
|---|---|
| Representation | 73.8 |
| Embedding | **74.2** |

## 6 DISCUSSION AND CONCLUSION

We formalize Supervised Representation Learning (SRL) from an information-theoretic perspective and propose Supervised Dimension Contrastive Learning (SupDCL) to optimize it. Existing methods have limitations - SupCon suffers from limited diversity, while SSL lacks semantic alignment. SupDCL addresses this by jointly optimizing representation entropy, residual information, and prediction uncertainty using dedicated losses. Experiments show that SupDCL achieves superior in- and out-domain performance, highlighting its ability to learn transferable representations.

**Limitations**  Despite these successes, SupDCL relies on using fixed-form class labels for supervision. This limitation suggests potential avenues for future work. Expanding SupDCL to handle more diverse forms of supervision, such as missing or noisy labels or even non-categorical and multi-task labels, could further enhance its generalizability.

**Broader Impacts**  The proposed SupDCL framework can benefit diverse applications by learning rich, transferable representations that balance diversity and task-relevance. The theoretical insights can also inspire further advancements in representation learning.

## REPRODUCIBILITY STATEMENT

We have taken several steps to ensure the reproducibility of our work. The formulation of Supervised Dimension Contrastive Learning (SupDCL) is presented in Section 4, with each loss component clearly defined. Implementation specifics, including optimizer settings, learning rate schedules, batch sizes, and training epochs for both ImageNet-1K pretraining and CIFAR-100 ablation studies, are described in Section 5 and detailed further in Appendix D. Additional analyses, such as computational cost evaluation (Appendix C) and intrinsic dimensionality estimation (Appendix A), enhance transparency. Comprehensive ablation studies (Section 5.3) isolate the contribution of each component. For baselines, we rely on official implementations or publicly released checkpoints provided by the respective authors, ensuring faithful comparisons. Dataset usage (ImageNet-1K and 10 transfer benchmarks) is clearly described with appropriate references. To further facilitate reproducibility, we will release the source code, enabling researchers to reproduce experimental results.

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

# A  ADDITIONAL DIVERSITY MEASURES

To complement the PCA-based intrinsic dimensionality (PC-ID) analysis in the main text, we further evaluate the diversity of learned representations using alternative metrics that capture structural aspects of entropy. This section reports results based on (i) spectral entropy of the eigenvalue distribution, (ii) log-determinant of the normalized covariance spectrum, and (iii) intrinsic dimensionality estimated by the TwoNN method. These measures provide complementary perspectives on representation dispersion beyond the Gaussian setting assumed by PC-ID.

## A.1  SPECTRAL ENTROPY AND LOG-DETERMINANT

Given the covariance eigenvalues $\{\lambda_i\}$, we compute the spectral entropy (Panda et al., 2023):

$$H_{\text{spec}} = -\sum_i \tilde{\lambda}_i \log \tilde{\lambda}_i, \quad \tilde{\lambda}_i = \frac{\lambda_i}{\sum_j \lambda_j}, \tag{A1}$$

which quantifies the flatness of the eigen-spectrum. A flatter spectrum yields higher entropy, indicating more uniformly dispersed variance.

We also report the log-determinant of the normalized covariance spectrum:

$$\log \det \tilde{\Sigma}_Y, \tag{A2}$$

which is closely related to Gaussian entropy in a scale-invariant manner.

Table A1: Comparison of PC-ID, spectral entropy, and log-determinant across methods. Higher values indicate greater representation diversity.

| Method | PC-ID | Spectral Entropy | Log-determinant |
|---|---|---|---|
| Supervised | 0.22 | 5.53 | -2.03e4 |
| SupCon | 0.61 | 6.45 | -1.79e4 |
| Barlow Twins | 0.75 | 6.72 | -1.71e4 |
| SupDCL | 0.78 | 6.84 | -1.69e4 |

These results demonstrate that although $\hat{H}(Y)$ is not a direct estimate of entropy, the ordering across methods is consistent with spectral entropy and log-determinant, supporting the interpretation of PC-ID as a structure-sensitive entropy proxy.

## A.2  INTRINSIC DIMENSIONALITY VIA TWONN

Since real-world learned representations need not follow Gaussian assumptions and may reside on nonlinear manifolds, we also estimate intrinsic dimensionality using the TwoNN method (Facco et al., 2017). TwoNN exploits local neighborhood statistics to approximate the number of active degrees of freedom in the representation, providing a nonparametric proxy for entropy.

Table A2: Intrinsic dimensionality estimated by TwoNN across methods. Higher values indicate larger effective degrees of freedom.

| Method | TwoNN Intrinsic Dim. |
|---|---|
| Supervised | 12.81 |
| SupCon | 12.96 |
| Barlow Twins | 13.79 |
| SupDCL | 14.64 |

The ordering is consistent with PC-ID and other proxies: models with higher intrinsic dimensionality also achieve higher out-domain accuracy, reinforcing the conclusion that greater dispersion in representations supports generalization.

# B MUTUAL INFORMATION DECOMPOSITION

**Proof of Proposition 4.1**  We start from the definition of mutual information:

$$I(Y;C) = H(Y) - H(Y|C).$$

Since the representation $Y$ can be decomposed as $Y = Y_{\text{range}} \oplus Y_{\text{null}}$, the conditional entropy $H(Y|C)$ can be rewritten using the chain rule of entropy:

$$H(Y|C) = H(Y_{\text{range}}, Y_{\text{null}}|C) = H(Y_{\text{range}}|C) + H(Y_{\text{null}}|Y_{\text{range}}, C).$$

The aggregator $h$ is defined as $h(Y) = W_h Y = \hat{C}$, and since $W_h Y_{\text{null}} = 0$, it follows that

$$\hat{C} = W_h Y_{\text{range}}.$$

Because $W_h$ is bijective on $\text{Row}(W_h)$ and $Y_{\text{range}} \in \text{Row}(W_h)$, $Y_{\text{range}}$ is fully determined by $\hat{C}$. Thus, the conditional entropy $H(Y_{\text{range}}|C)$ can be expressed as:

$$H(Y_{\text{range}}|C) = H(\hat{C}|C).$$

Moreover, since $W_h$ is bijective on $\text{Row}(W_h)$, we can replace $Y_{\text{range}}$ with $\hat{C}$ in the conditional entropy of $Y_{\text{null}}$, leading to:

$$H(Y_{\text{null}}|Y_{\text{range}}, C) = H(Y_{\text{null}}|\hat{C}, C).$$

Substituting these results into the equation for $H(Y|C)$, we get:

$$H(Y|C) = H(\hat{C}|C) + H(Y_{\text{null}}|\hat{C}, C).$$

Rearranging the mutual information formula, we have:

$$I(Y;C) = H(Y) - H(Y|C),$$

which becomes:

$$I(Y;C) = H(Y) - H(\hat{C}|C) - H(Y_{\text{null}}|\hat{C}, C).$$

Thus, we conclude:

$$I(Y;C) = H(Y) - H(Y_{\text{null}}|\hat{C}, C) - H(\hat{C}|C).$$

This completes the proof.  $\square$

## C    COMPUTATIONAL COST ANALYSIS

To assess scalability and computational efficiency, we report the per-sample forward and backward FLOPs for our method using a 3-layer MLP projector, computed across different ResNet backbones. Note that the orthogonal loss is applied solely on the lightweight aggregator network (a fixed-size 3-layer MLP), and thus contributes negligible computational overhead. Consequently, it is excluded from the reported FLOPs.

The results are summarized in Table A3. We compare the base method Barlow Twins with our proposed SupDCL. The overhead in SupDCL mainly arises from the class correlation loss, which is independent of backbone complexity and becomes relatively less significant as the model size increases.

Table A3: **Computational Costs.** Per-sample forward and backward FLOPs (in GFLOPs) for different ResNet backbones. SupDCL introduces a small overhead over Barlow Twins, primarily due to the class correlation loss.

| Method | ResNet-50 | ResNet-101 | ResNet-152 |
|---|---|---|---|
| Barlow Twins | 18.59 | 33.43 | 48.28 |
| SupDCL | 20.28 (**+9.1%**) | 35.15 (**+5.2%**) | 50.02 (**+3.6%**) |

This modest increase in computational cost is justified by the representational improvements introduced by SupDCL. We also note that representation learning methods often require long training schedules (typically 800–1000 epochs) to achieve strong generalization.

## D    IMPLEMENTATION DETAIL

### D.1    IN-DOMAIN CLASSIFICATION

For the linear evaluation protocol used after pretraining on ImageNet-1K, a standard linear classifier is trained on top of frozen features. Specifically, the representation is first extracted from the pretrained model, and then a linear layer is trained on the frozen features to classify ImageNet-1K data. During training, common augmentations such as random resized crops and horizontal flips are applied. The accuracy is reported on the central crop of each validation image. The model is optimized for 100 epochs using stochastic gradient descent (SGD) with a momentum of 0.9, a batch size of 128 and a base learning rate of 0.01.

### D.2    OUT-DOMAIN TRANSFER LEARNING

We perform transfer learning with the linear evaluation on 10 datasets: CIFAR10/100 (Krizhevsky et al., 2009), Food (Bossard et al., 2014), Pets (Parkhi et al., 2012), Flowers (Nilsback & Zisserman, 2008), Caltech101 (Fei-Fei et al., 2004), Cars (Krause et al., 2013), Aircraft (Maji et al., 2013), DTD (Cimpoi et al., 2014), SUN397 (Xiao et al., 2010). We follow the standard linear transfer evaluation protocol (Sun et al., 2017), training linear classifiers on frozen features from $224 \times 224$ resized images without data augmentation. We use L-BFGS to minimize $\ell_2$-regularized cross-entropy loss, selecting the regularization parameter from 45 logarithmically spaced values between $10^{-6}$ and $10^5$ via the validation set. Then, the linear classifier is retrained on both training and validation data with test accuracy.

## E    THE USE OF LARGE LANGUAGE MODELS (LLMS)

During the preparation of this paper, large language models were employed in a restricted and supervised manner. Their role was restricted to English proofreading and grammar correction. They were not employed to draft complete sentences or passages, nor to generate novel methods or results.

