# OpenReview forum: "Supervised Dimension Contrastive Learning"
_ICLR.cc/2026/Conference — ICLR 2026 Conference Withdrawn Submission_

### Official Review · Reviewer_kCWF · 2025-10-29

**Soundness:** 2
**Presentation:** 2
**Contribution:** 2
**Rating:** 2
**Confidence:** 4

**Summary:**

The paper introduces a new objective function that comprise with decorrelation, orthogonal, and class correlation part each optimizing $H(Y)$, $H(Y_{null}|\hat{C},C)$, and $H(\hat{C}|C)$, respectively. This decomposition can balance pros and cons in self-supervised representation learning and supervised representation learning, keep representation power on unseen domain while keep keep class-correlation. The method is evaluated on In-domain and out-domain image top-1 accuracy.

**Strengths:**

- The decomposition of mutual information could be interesting.
- The paper is well written.

**Weaknesses:**

- The empirical performance is weak. The paper says it captures benefits of both self-supervised methods and supervised methods, but to me, it seems SupDCL captures weakness of both methods. In distribution performance is worse then supervised methods, while out-distribution performance is worse then self-supervised methods. If I am a real user, I would choose either self-supervised model or supervised model depending on my problem setting, not SupDCL.
- The method combines three entropy terms, and optimizing each term independently. I  assume balancing these terms is extremely difficult (authors used 0.04 and 0.02 for decorrelation, which implies they did some grid search) and highly sensitive. The paper contains sensitivity analysis in Figure 4, but I would expect authors to provide some rule how to choose optimal hyperparameters.

**Questions:**

- How do you think about the weakness 1? In which case SupDCL is beneficial than self-supervised or supervised models?
- Can you provide more comprehensive sensitivity analysis? or any suggestions how to choose optimal hyperparameters?

---

### Official Review · Reviewer_ccve · 2025-10-31

**Soundness:** 3
**Presentation:** 3
**Contribution:** 3
**Rating:** 4
**Confidence:** 3

**Summary:**

This paper addresses a well-documented trade-off in visual representation learning: the tension between the high out-of-domain generalization of self-supervised learning methods, which learn diverse features, and the strong in-domain performance of supervised representation learning methods, which excel at class-separability but often at the cost of feature diversity.
The core contribution is a principled reframing of the SRL problem through the lens of information theory. The authors propose that the objective of SRL should be to maximize the mutual information $I(Y;C)$ between the learned representations $Y$ and the supervision signal (class labels) $C$. Building on this framework, the paper introduces Supervised Dimension Contrastive Learning (SupDCL), and the authors assert that their method achieves a state-of-the-art balance, bridging the performance gap between leading SSL and SRL paradigms.

**Strengths:**

1. Novel framing. The paper reconceptualizes supervised representation learning as mutual-information maximization between features and labels. This yields a principled decomposition that motivates the three losses, which is a strong conceptual contribution.
2. Comprehensive objective. SupDCL integrates ideas from SSL (Barlow Twins style decorrelation and invariance) with supervision. The decorrelation and orthogonality losses clearly promote richer, less redundant features, while the class-correlation loss enforces alignment with labels.

**Weaknesses:**

1. State-of-the-Art Claims:
The claims of achieving "state-of-the-art generalization" are unsubstantiated due to the complete omission of modern and dominant baselines from the MIM and JEPA paradigms (e.g., MAE, I-JEPA). The authors' justification for excluding these methods—being "restricted to ResNet-50 backbones due to computational limits"—is insufficient for a paper submitted to ICLR that makes such strong claims. The omission is critical because these newer paradigms represent a conceptual shift away from the contrastive/non-contrastive methods that form the entirety of the paper's SSL baselines. Without a comparison to these methods, it is impossible to assess whether SupDCL is truly advancing the frontier of representation learning or merely achieving a local optimum within an older, potentially superseded, paradigm.

2. Theory-Practice Gap:
There is a fundamental disconnect between the linear theory used to derive the method's objectives and the non-linear model required for the reported performance. This undermines the paper's central claim of providing a "principled" framework. The information-theoretic decomposition in Proposition 4.1, and its proof in Appendix B, hinges on the assumption that $h$ is a linear operator, i.e., $h(Y) = W_h Y$. This linearity is what permits the clean decomposition of the representation space $Y$ into the range and null spaces of $W_h$ via its pseudo-inverse, $W_h^\dagger$. However, the experiments use a 3-layer non-linear MLP for the aggregator, which the authors' ablation study (Table 3) shows is critical for achieving the best performance. For a non-linear function like an MLP, the concepts of a single weight matrix $W_h$, its pseudo-inverse, and a well-defined null space do not hold. While orthogonal regularization is a valid technique for MLPs, its benefits are typically understood in terms of improving gradient flow and reducing parameter redundancy, not cleanly minimizing a specific entropy term as claimed. This gap undermines the paper's central claim of providing a "principled framework."

**Questions:**

Related to the above "Weaknesses", the questions are:
1. Could you provide a more thorough justification for the exclusion of all MIM and predictive architecture baselines? At a minimum, could you include a detailed discussion of how SupDCL's information-theoretic philosophy compares to these paradigms and provide hypotheses on its expected relative performance?
2. How does your theoretical decomposition extend to the non-linear MLP aggregator used in practice? Can you provide a theoretical argument or at least a strong intuition for why the three loss components still correspond to optimizing the terms of the mutual information objective in the non-linear case?

---

### Official Review · Reviewer_pGA1 · 2025-11-01

**Soundness:** 4
**Presentation:** 4
**Contribution:** 3
**Rating:** 6
**Confidence:** 3

**Summary:**

The paper introduces a supervised representation learning method built on a mutual information perspective. The approach combines three objectives: a decorrelation loss between feature dimensions, an orthogonality loss on the mapper weights to reduce redundancy, and a discriminative alignment loss to align predictions with class labels. The authors show strong results on ImageNet-1K, achieving better in-domain and out-of-domain accuracy than baselines such as SupCon, DINO, and cross-entropy.

**Strengths:**

1- the proposed method outperforms strong baselines such as SupCon, DINO, and cross-entropy on both in-domain and out-of-domain accuracy on ImageNet-1K.

2- the orthogonality loss seems to be a novel design choice relative to prior work as it operates on the mapper’s weights rather than solely on the embeddings.

3- the paper is clearly written and easy to follow, with sufficient implementation details to support reproducibility.

**Weaknesses:**

1- the proposed loss looks like an extension of the Barlow Twins loss, mainly differentiated by the orthogonality term. It would help to highlight more clearly what conceptual or empirical advantages this formulation offers beyond Barlow Twins.

2- it would be good to include a large and diverse dataset for out-of-distribution evaluation, such as Places365, since most of the current OOD datasets are relatively small, have limited image sizes, or few classes.

**Questions:**

1- in Figure 6 and 461-470,  the authors discuss an in-domain/out-of-domain accuracy trade-off for SupDCL as the number of dimensions increases, but this behavior doesn’t appear for Barlow Twins. Could you elaborate on this difference or provide a theoretical explanation for the trade-off?

2- the theoretical analysis assumes a linear aggregator, but no results are shown for a linear version. Could you include results for that case and discuss whether they match the theoretical expectations?

3- can you extend the ablations in Table 2 to analyze the performance contribution of the two main loss components separately?

4- how does this framework compare to other information-theoretic representation learning frameworks that aim for unification, such as:

[1] Balestriero, Randall, and Yann LeCun. "Contrastive and non-contrastive self-supervised learning recover global and local spectral embedding methods." NeurIPS 2022

[2] Tschannen, Michael, et al. "On mutual information maximization for representation learning."ICLR 2020

[3] Alshammari, Shaden, et al. "I-Con: A unifying framework for representation learning." ICLR 2025

[4] Sobal, Vlad, et al. "X-Sample Contrastive Loss: Improving Contrastive Learning with Sample Similarity Graphs." ICLR 2024.

minor suggestion: can you add a short pseudocode snippet summarizing the final objective function and training loop? It would help clarify how the three loss components interact in practice. Also, please comment on whether there is any computational overhead compared to other methods like SupCon or Barlow Twins.

---

### Official Review · Reviewer_Yffc · 2025-11-08

**Soundness:** 2
**Presentation:** 2
**Contribution:** 2
**Rating:** 2
**Confidence:** 4

**Summary:**

This paper presents a framework for supervised representation learning grounded in information theory. The key idea is to balance representation diversity (maximizing H(Y)) and class relevance (minimizing H(Y∣C)) by optimizing their mutual information I(Y;C). To this end, the authors decompose the representation space via a learnable linear aggregator hand propose three complementary loss terms: (1) decorrelation loss to enhance feature independence and diversity, (2) orthogonal loss to minimize redundancy by enforcing orthogonality in the aggregator, and (3) class correlation loss to align predicted and true class distributions. Experiments on ImageNet-1K and ten downstream datasets show that SupDCL bridges the gap between self-supervised and supervised learning, achieving competitive or better performance than baselines such as SupCon, Barlow Twins, and DINO while maintaining reasonable computational cost.

**Strengths:**

- The authors provide clear mapping from theory to implementation: each loss corresponds to a distinct mutual information term.
- The proposed framework achieves strong empirical results showing balance between in-domain and out-domain generalization.

**Weaknesses:**

- Limited exploration of modern architectures (ViTs, masked modeling baselines) restricts relevance to current large-scale settings.
- “State-of-the-art” claim is overstated given omission of stronger 2023–2025 baselines like DINOv2/v3 and I-JEPA.
- The proposed “class correlation loss” is conceptually similar to correlation-based label alignment used in prior VICReg-style works, offering modest originality.
- Mutual information motivation is largely heuristic—no quantitative MI estimation or validation of entropy proxies.
- Experimental results focus mostly on linear evaluation; finetuning or semi-supervised adaptation tests would strengthen the case for transferability.
- Some sections (e.g., orthogonal loss derivation) could benefit from clearer intuition and visualization.

**Questions:**

- Could the authors provide direct empirical evidence that the estimated entropy proxies (PC-ID, TwoNN, spectral entropy) correlate with I(Y;C)?
- How sensitive is performance to the assumption of linear aggregator h? Would nonlinear hbreak the MI decomposition?
- Does SupDCL scale to ViT or larger datasets beyond ImageNet-1K without loss of stability?
- Can the proposed orthogonal loss conflict with batch normalization or weight decay in practice?
- Would integrating SupDCL with masked image modeling (e.g., DINOv2) further improve transfer learning?

**Details Of Ethics Concerns:**

No ethics concerns.

---

### Note · Authors · 2025-11-26

**Comment:**

We would like to withdraw our rebuttal. Thank you to all reviewers for your valuable time and constructive feedback.

**Withdrawal Confirmation:**

I have read and agree with the venue's withdrawal policy on behalf of myself and my co-authors.